# Association between serum lipids concentration and patients with age-related cataract in China: a cross-sectional, case–control study

Shengjie Li,[1,2] Danhui Li,[3] Yudong Zhang,[1] Jisen Teng,[1] Mingxi Shao,[1] Wenjun Cao[1,2]

MS and WC contributed equally.

[1]Department of Clinical Laboratory, Eye and ENT Hospital, Shanghai Medical College, Fudan University, Shanghai, China
[2]Department of Ophthalmology and Visual Science, Eye and ENT Hospital, Shanghai Medical College, Fudan University, Shanghai, China
[3]Key Laboratory of Environment and Genes Related to Diseases, Ministry of Education, Medical School, Xi'an Jiaotong University, Xi'an, China

**Correspondence to**
Dr. Mingxi Shao;
smingxi@sohu.com and Dr Wenjun Cao;
wgkjyk@aliyun.com

## ABSTRACT

**Objective** Obesity and a high-fat diet have been found to be associated with an increased risk of age-related cataract (ARC). Thus, this study aimed to investigate whether serum lipid levels are associated with the incidence of ARC.

**Design** Cross-sectional, case–control study.

**Setting** EyeandENT Hospital of Fudan University, Shanghai, China.

**Participants** A total of 219 ARC (male=94, female=125) subjects and 218 (male=110, female=118) normal control subjects were recruited in this study.

**Outcome measures** A detailed eye and systematic examination was performed. Serum high-density lipoprotein cholesterol (HDL-C), low-density lipoprotein cholesterol (LDL-C), triglyceride (TG) and cholesterol (CHO) levels were measured by enzymatic colorimetry, and serum apolipoprotein A (APOA) and apoB (APOB) levels were measured by immunoturbidimetry. The subgroups were classified according to gender and types of disease (cortical, nuclear and posterior subcapsular cataract). Logistic regression analyses were performed to identify the association between serum lipid levels and ARC.

**Results** The serum LDL-C, TG, CHO and APOA levels were significantly higher (p<0.05) in the ARC group than in the control group. A similar result was observed when the serum lipid concentrations were compared between the ARC and control groups both in male and female subgroups. A higher proportion of individuals in the ARC group had higher LDL-C, TG, CHO and APOA levels (fold=3.45, 17.37, 3.27 and11.91, respectively; p<0.0001 in all cases) than in the control group. Results of the logistic regression analyses revealed that high LDL-C (ORs=1.897, 95% CI 0.960 to 3.678) and TG (OR=1.854, 95% CI 1.232 to 2.791) were the independent risk factors for ARC.

**Conclusion** The serum LDL-C and TG levels were demonstrated to be independent risk factors for ARC.

## Strengths and limitations of this study

► This study provides evidence for a better understanding on the association between serum lipids concentration and patients with age-related cataract.
► The detailed information of demographic and clinical data were recquired.
► The subjects involved in the study are from a racially homogeneous background (all Chinese).
► We did not consider treatment with lipid-lowering medications and how it might affect the results.

There are three types of ARC based on the location of the opacity within the lens: cortical, nuclear and posterior subcapsular (PSC).[3] Surgery, which is the only available treatment for ARC, is a major medical cost given China's rapidly ageing populations and large numbers of ARC patients. ARC is a complex disease with multiple risk components, including high myopia,[2] diabetes,[4] hyperglycaemia,[5] obesity,[6] blood pressure[4] and lipids.[7] Among these risk factors, obesity, blood pressure and lipids have been found to be consistently associated with blood lipid concentration.

Whether elevated blood lipid concentration is a risk factor for ARC is uncertain. Hiller et al[7] reported that fasting hypertriglyceridaemia (≥250 mg/dL) is associated with an increased risk of PSC cataract in men. In a population-based, cross-sectional study, Tang et al[2] found that high low-density lipoprotein (LDL) and low high-density lipoprotein (HDL) were independent risk factors for ARC. However, Marks et al[8] reported that both men and women with cataracts had significantly lower serum cholesterol (CHO) concentrations than subjects without cataracts. The relationship between blood lipid concentration and ARC risk is less consistent.

## INTRODUCTION

Age-related cataract (ARC), one of the leading causes of blindness and poor vision worldwide,[1] remains a severe public health challenge worldwide, particularly in China.[2]

Some have suggested that blood lipid concentration is an increased risk factor for ARC,[5 9 10] and others have shown the opposite effect[8 11] or no association.[12] Here, we conducted a large, cross-sectional, case–control study to detect and compare the blood lipid levels between ARC patients and normal subjects and to investigate the possible association between blood lipid levels and the risk of ARC.

## METHODS
### Patient and public involvement statement
This study was approved by the Ethics Committee of the Eye and ENT Hospital of Fudan University (EENT2015011), Shanghai and complied with the Declaration of Helsinki. Informed consent was obtained from all the subjects. Patients with ARC were consecutively recruited from the Department of Ophthalmology and Visual Science, Eye and ENT Hospital of Fudan University between June 2016 and June 2017. Both newly diagnosed and referral ARC patients were included. Normal controls were consecutively recruited from individuals who participated in annual health screenings during the study period. The study clerks conducted face-to-face questionnaire interviews to gather data such as life habits, tobacco and alcohol intake, height and body weight. The subjects were included according to inclusion criteria.

Medical examinations included assessments of ECGs, X-rays, liver function, renal function, infectious disease, blood pressure, heart rate and body temperature. All subjects underwent medical examinations at the Eye and ENT Hospital of Fudan University. Alcohol consumption (>3 times per week for more than 6 months), smoking (>1 cigarette per day and more than 6 months (current or former)) (self-reported) and dietary habits (meat (less vegetables)/vegetables (less meat)) were also collected. Furthermore, each ARC patient underwent a standardised ophthalmic examination performed by cataract specialists. A visual acuity measurement was obtained for each patient based on the International Standard Visual Acuity Chart. The details of the standardised ophthalmic examination have been previously described.[13]

### Inclusion criteria
#### ARC subjects
A total of 320 cataract subjects were recruited, of whom 101 (metabolic cataract=15, traumatic cataract=8, congenital cataract=17, secondary cataract=34, cancer=3, renal disease=3, liver disease=5, other eye diseases=16) were excluded from the study based on the inclusion criteria. If both eyes suffered from ARC, only one eye was randomly selected for the study. The final sample consisted of 219 ARC patients who met the following inclusion criteria as described previously[13]: age ≥45 years; ARC recorded using the Lens Opacities Classification System III; ARC defined as the presence of any type of ARC (cortical, nuclear or PSC) or a history of cataract surgery (pseudophakic or aphakic eyes) in either eye; no history of secondary cataracts, congenital cataracts or any other eye diseases and no history of systemic diseases such as acute infectious diseases, kidney disease, autoimmune disease and cancer.

### Control subjects
A total of 250 subjects were recruited. Of these, 32 normal subjects (cataract=5, cancer=2, renal disease=6, liver disease=4, autoimmune disease=2, macular degeneration=3, cardiovascular disease=4, other diseases=6) were later excluded from the study based on the inclusion criteria. The final sample consisted of 218 control subjects who met the following previously described criteria[13]: age ≥45 years; no history of any types of cataracts, secondary cataracts, congenital cataracts and any other eye diseases and no history of systemic diseases such as acute infectious diseases, kidney disease, autoimmune disease and cancer.

### Data collection
Clinical and demographic information were obtained from the medical data platform of Eye and ENT Hospital by trained staff (Shengjie Li and Mingxi Shao) using standardised data collection and quality control procedures, which produced reliable data for analysis.

For serum lipid measurements, blood samples were obtained in the morning after the subjects had fasted for 8 hours via standard venipuncture of the veins in the antecubital fossae. Blood coagulation was prevented using EDTA, and the obtained samples were centrifuged within 60 min (3000 rpm, 10 min, 20°C). Serum HDL cholesterol (HDL-C), LDL cholesterol (LDL-C), triglyceride (TG) and CHO levels were measured by enzymatic colourimetry using a commercially available kit (Roche Diagnostics GmbH, Mannheim, Germany). Serum apolipoprotein A (APOA) and B (APOB) levels were measured by immunoturbidimetry using a commercially available kit (Roche Diagnostics, GmbH, Mannheim, Germany). Internal controls were analysed daily over a 2-year period, with typical monthly coefficient of variation of 2%–5% and no significant changes in the values.

### Statistical analyses
The results of this study are presented as mean±SD deviation. Normality of the data was assessed with the Kolmogorov-Smirnoff test. Independent student's t-test and $\chi^2$ test were used to compare the subject characteristics between the ARC and control groups. One-way analysis of variance was used to compare the serum lipid levels among the three groups (cortical, nuclear and PSC). Based on the normal reference range value of LDL-C (LDL-C ≥3.36 mmol/L, LDL-C <3.36 mmol/L), TG (TG ≥2.26 mmol/L, TG <2.26 mmol/L), CHO (CHO ≥5.20 mmol/L, CHO <5.20 mmol/L) and APOA (APOA ≥1.9 mmol/L, APOA <1.9 mmol/L), the ARC and control subjects were divided into lower and higher groups, respectively. The prevalence of ARC in China is higher in women than in men (37.2% in men and

**Table 1** Demographics of subjects between ARC and control group

|  | ARC group (n=219) | Control group (n=218) | t value | P value |
|---|---|---|---|---|
| Age (year) | 66.99±8.83 (46-85) | 66.00±7.10 (48-79) | 1.281 | 0.201 |
| Gender (male/female) | 94/125 | 110/118 | 1.276 | 0.259 |
| BMI (kg/m$^2$) | 23.68±3.16 | 21.55±2.74 | 7.476 | <0.001 |
| SBP (mm Hg) | 132.45±15.50 | 129.92±5.55 | 2.230 | 0.027 |
| DBP (mm Hg) | 74.21±10.65 | 76.58±5.37 | 2.896 | 0.004 |
| IOP (mm Hg) | 13.84±3.59 | – | – | – |
| Visual acuity | 0.23±0.15 | – | – | – |
| Smoking (yes/no) | 15/204 | 18/200 | 0.310 | 0.578 |
| Drinking (yes/no) | 33/186 | 23/195 | 1.996 | 0.158 |
| Diet (meat/vegetarian) | 159/60 | 98/120 | 34.476 | <0.001 |

Data are expressed as mean ±SD deviation (SD). Independent-Samples T Test and $\chi^2$ test was used.
ARC, age-related cataract; BMI, body mass index; DBP, diastolic blood pressure; IOP, intraocular pressure; SBP, systolic blood pressure.

39.0% in women).[2] Moreover, in our study, there were more female ARC subjects than male ARC subjects (125 vs 94). Therefore, the subjects were categorised into gender subgroups. A two-sided p<0.05 was considered statistically significant. Logistic regression analyses were performed to identify the association between serum lipid levels and ARC (control group=0, ARC=1; male=0, female=1). ORs with 95% CI were estimated by logistic regression analyses. All analyses were performed using SPSS 13.0 software.

## RESULTS

A total of 219 ARC (male=94, female=125) subjects and 218 (male=110, female=118) normal control subjects were recruited in this study. There was no significant difference in the mean age, gender, smoking and drinking between the ARC and control groups (p>0.005). However, there was a significant difference in the body mass index (BMI), systolic blood pressure, diastolic blood pressure and diet between the ARC and control groups (p<0.05). Table 1 presents a summary of the demographics of the ARC and control groups. Online supplementary figure 1 presents a summary of the correlations of individual subject's age with lipid profile.

### Comparison of serum HDL-C, LDL-C, TG, CHO, APOA and APOB levels between the ARC and control subjects

Table 2 shows the HDL-C, LDL-C, TG, CHO, APOA and APOB levels in the ARC group. The level of serum LDL-C, TG, CHO and APOA was significantly higher (p<0.05) in the ARC group than in the control group.

Moreover, the ARC and control subjects were categorised into female and male subgroups. In both the female and male subgroups, the TG level was significantly higher (p<0.05) in the ARC group than in the control group. In the male subgroup, the APOA level was significantly higher (p<0.05) in the ARC group than in the control group. In the female subgroup, the level of LDL-C was significantly higher (p<0.05) in the ARC group than in the control group.

### Serum LDL-C, TG, CHO and APOA levels in the ARC and control groups

The ARC and control subjects were each divided into a lower group and a higher group according to whether their serum LDL-C, TG, CHO and APOA values were below or above the normal reference range, respectively (table 3). Compared with the control group, more subjects from the ARC group were in the higher group (3.45, 17.37, 3.27 and 11.91-fold, respectively, p<0.0001).

### Comparison of serum lipid levels in subjects with different types of ARC

Of the 219 ARC patients, 85 had nuclear cataracts, 108 had cortical cataracts and 26 had PSC. Table 4 presents a comparison of the serum lipid levels and the demographic parameters in the ARC subjects. There were no statistically significant differences in the demographic and serum lipid levels among the nuclear, cortical and PSC cataract groups (p>0.05).

### Logistic regression analysis of the association between serum lipid levels and ARC

Logistic regression analyses were performed to identify the association between serum HDL-C, LDL-C, TG, CHO, APOA and APOB levels and ARC in comparison to subjects without ARC (table 5). Logistic regression analyses revealed that being female (OR=1.719, 95% CI 1.074 to 2.752) and BMI (OR=1.297, 95% CI 1.194 to 1.408), systolic blood pressure (OR=1.029, 95% CI 1.008 to 1.051), LDL-C (OR=1.897, 95% CI 0.960 to 3.678) and TG levels (OR=1.854, 95% CI 1.232 to 2.791) were associated with ARC after adjusting for age and other demographic parameters.

## DISCUSSION

In this case–control study of the Chinese Han population, we found the following: First, the serum LDL-C, TG,

**Table 2** Comparison of serum lipid levels between ARC and control group, stratified according to gender

|  | ARC group | Control group | t value | P value |
|---|---|---|---|---|
| LDL-C (mmol/L) | 2.65±0.93 | 2.46±0.64 | 2.461 | 0.014 |
| Male | 2.45±0.85 | 2.43±0.60 | 0.184 | 0.854 |
| Female | 2.82±0.97 | 2.49±0.67 | 2.894 | 0.004 |
| TG (mmol/L) | 1.56±1.25 | 1.18±0.35 | 4.238 | <0.001 |
| Male | 1.54±1.42 | 1.15±0.36 | 2.609 | 0.010 |
| Female | 1.57±1.10 | 1.22±0.35 | 3.364 | 0.001 |
| HDL-C (mmol/L) | 1.14±0.43 | 1.19±0.28 | 1.201 | 0.231 |
| Male | 1.10±0.48 | 1.10±0.28 | 0.064 | 0.949 |
| Female | 1.18±0.39 | 1.28±0.25 | 2.158 | 0.032 |
| APOA (mmol/L) | 1.34±0.23 | 1.28±0.36 | 2.270 | 0.024 |
| Male | 1.30±0.23 | 1.22±0.36 | 1.983 | 0.048 |
| Female | 1.39±0.22 | 1.33±0.35 | 1.649 | 0.101 |
| APOB (mmol/L) | 0.85±0.26 | 0.86±0.19 | 0.772 | 0.441 |
| Male | 0.81±0.25 | 0.83±0.18 | 0.757 | 0.450 |
| Female | 0.87±0.26 | 0.89±0.19 | 0.590 | 0.556 |
| CHO (mmol/L) | 4.47±1.78 | 4.26±0.79 | 2.146 | 0.032 |
| Male | 4.21±1.10 | 4.03±0.69 | 1.438 | 0.152 |
| Female | 4.67±1.20 | 4.49±0.81 | 1.275 | 0.204 |

Data are expressed as mean±SD deviation. Independent samples t-test was used.
APOA, apolipoprotein A; APOB, apolipoprotein B; ARC, age-related cataract; CHO, cholesterol; HDL-C, high-density lipoprotein cholesterol; LDL-C, low-density lipoprotein cholesterol; TG, triglyceride.

CHO and APOA levels was significantly higher (p<0.05) in the ARC group than in the control group. Second, the proportion of individuals with higher LDL-C, TG, CHO and APOA levels was higher in the ARC group (3.45, 17.37, 3.27 and 11.91-fold, p<0.0001) relative to the control group. Finally, high serum LDL-C (OR=1.897, 95% CI 0.960 to 3.678) and TG levels (OR=1.854, 95% CI 1.232 to 2.791) were associated with ARC after adjusting for age, gender and other demographic parameters. The results of this case–control study indicate that higher serum LDL-C and TG levels are associated with a significantly increased risk of ARC.

In a study on the association of metabolic syndrome and ARC, reduced HDL-C and elevated TG levels

**Table 3** The number of subjects in ARC and control group, according to LDL-C, TG, CHO and APOA levels, respectively

|  | ARC group | Control group | Fold (t value) | P value |
|---|---|---|---|---|
| LDL-C (mmol/L) |  |  |  |  |
| <3.36 | 167 | 203 |  |  |
| ≥3.36 | 52 (52/219=23.74%) | 15 (15/218=6.88%) | 3.45 (17.818) | <0.001 |
| TG (mmol/L) |  |  |  |  |
| <2.26 | 184 | 216 |  |  |
| ≥2.26 | 35 (35/219=15.98%) | 2 (2/218=0.92%) | 17.37 (27.136) | <0.001 |
| CHO (mmol/L) |  |  |  |  |
| <5.20 | 153 | 198 |  |  |
| ≥5.20 | 66 (66/219=30.14%) | 20 (20/218=9.17%) | 3.27 (20.549) | <0.001 |
| APOA (mmol/L) |  |  |  |  |
| <1.9 | 207 | 217 |  |  |
| ≥1.9 | 12 (12/219=5.48%) | 1 (1/218=0.46%) | 11.91 (8.996) | <0.001 |

$\chi^2$ test was used.
APOA, apolipoprotein A; ARC, age-related cataract; CHO, cholesterol; LDL-C, low-density lipoprotein cholesterol; TG, triglyceride.

**Table 4** Comparison of serum lipid levels in subjects with different types of ARC

|  | Nuclear cataract | Cortical cataract | PSC | t value | P value |
|---|---|---|---|---|---|
| Age (year) | 67.54±8.83 | 66.59±8.55 | 66.87±10.39 | 0.282 | 0.754 |
| Gender (male/female) | 33/52 | 49/59 | 12/14 | 0.985 | 0.619 |
| BMI (kg/m$^2$) | 23.62±3.24 | 23.65±3.14 | 24.13±3.04 | 0.219 | 0.804 |
| SBP (mm Hg) | 134.28±16.65 | 131.15±14.38 | 131.63±16.61 | 1.006 | 0.367 |
| DBP (mm Hg) | 74.22±8.28 | 73.65±10.67 | 77.94±18.93 | 1.132 | 0.324 |
| LDL-C (mmol/L) | 2.67±0.97 | 2.60±0.87 | 2.81±1.08 | 0.491 | 0.612 |
| TG (mmol/L) | 1.46±1.28 | 1.68±1.32 | 1.38±0.64 | 1.022 | 0.362 |
| HDL-C (mmol/L) | 1.18±0.39 | 1.11±0.45 | 1.18±0.46 | 0.784 | 0.458 |
| APOA (mmol/L) | 1.29±0.34 | 1.27±0.38 | 1.29±0.35 | 0.107 | 0.899 |
| APOB (mmol/L) | 0.85±0.26 | 0.83±0.25 | 0.87±0.31 | 0.341 | 0.712 |
| CHO (mmol/L) | 4.45±1.22 | 4.47±1.11 | 4.51±1.38 | 0.020 | 0.981 |

Data are expressed as mean±SD. Independent samples t-test and $\chi^2$ test were used.
APOA, apolipoprotein A; APOB, apolipoprotein B; ARC, age-related cataract; BMI, body mass index; CHO, cholesterol; HDL-C, high-density lipoprotein cholesterol; LDL-C, low-density lipoprotein cholesterol; PSC, posterior subcapsular; SBP, systolic blood pressure; SDP, diastolic blood pressure; TG, triglyceride.

were found to be significantly associated with cataract in women.[14] In a population-based study (n=4926) in Beaver Dam, Klein et al[15] reported that higher serum HDL-C levels were associated with decreased risk of cataract. Moreover, an extremely strong association (p<0.0001) between HDL-C levels and the development of lens opacities was also reported in the South African population.[16] Furthermore, an elevated level of serum TG was significantly related to cataract in some studies.[7 17] In the present study, we also found that serum TG concentration was higher in the ARC group than in the normal group and that TG concentration was a risk factor of ARC, but the HDL-C level did not significantly differ between groups. Sabanayagam et al[4] also found no significant associations between serum HDL-C and cataract. The mechanism linking hyperlipidaemia and cataract is not clear. Thus far, no studies have clarified how changes in blood lipid affect the onset and/or development of ARC. Animal studies have

shown that hyperlipidaemia and low HDL-C are associated with an earlier onset and an elevated incidence of diabetic cataracts,[18] and inflammation and oxidative stress resulting from low HDL-C levels could induce cataract formation.[19] Moreover, it has been suggested that hyperlipidaemia is related to ARC through its associated complications such as cardiovascular disease.[9 15]

In the female and male ARC subgroups, the TG level was significantly higher (p<0.05) in the ARC group than in the control group. However, the APOA level was significantly higher in the ARC group than in the control group in the male subgroup but not in the female subgroup. In the female subgroup, the LDL-C level was significantly higher in the ARC group than in the control group but not in the male subgroup. Tavani et al[20] reported that hyperlipidaemia (OR, 1.8; 95% CI 1.2 to 2.7) was associated with an increased risk of cataract in women from northern Italy. In addition, elevated TG was also significantly associated with cataract in women but not in men.[14] A consistent gender difference in the association between blood lipid concentration and cataract was observed. Differences of hormonal[21] and lifestyle-related characteristics[22] between men and women may explain this gender difference.

In the subgroup analysis for ARC subtype, no statistically significant difference in the demographic and serum lipid levels was found among the nuclear, cortical and PSC cataract groups (p>0.05). Hiller et al[7] found that HDL-C levels were associated with PSC cataract but not in cortical and nuclear cataract. Moreover, Park et al[14] reported that reduced HDL-C was only significantly associated with nuclear cataract. The results of these existing studies are controversial, and the potential pathogenesis was unclear.

Our study also found that female gender (OR=1.719, 95% CI 1.074 to 2.752), high BMI (OR=1.297, 95% CI 1.194 to 1.408) and high systolic blood pressure

**Table 5** Logistic regression analysis of the association between serum lipid levels and ARC

|  | B | OR | P value | 95% CI |
|---|---|---|---|---|
| Female | 0.542 | 1.719 | 0.024 | 1.074 to 2.752 |
| BMI (kg/m$^2$) | 0.260 | 1.297 | <0.001 | 1.194 to 1.408 |
| SBP (mm Hg) | 0.029 | 1.029 | 0.006 | 1.008 to 1.051 |
| LDL-C (mmol/L) | 0.631 | 1.897 | 0.046 | 0.960 to 3.678 |
| TG (mmol/L) | 0.617 | 1.854 | 0.003 | 1.232 to 2.791 |

Logistic regression analyses were performed to identify the association between serum lipid levels and ARC (control group=0, ARC=1; male, 0, female=1) after adjusting for age, smoking, drinking, diastolic blood pressure, diet, high density lipoprotein cholesterol, APOA, APOB and cholesterol.
ARC, age-related cataract; BMI, body mass index; LDL-C, low-density lipoprotein cholesterol; SBP, systolic blood pressure; TG, triglyceride.

(OR=1.029, 95% CI 1.008 to 1.051), which were closely associated with serum lipid concentration, were associated with an increased risk of ARC after adjusting for age and other demographic parameters. In the Blue Mountains Eye Study, obesity (BMI ≥30 kg/m$^2$) and hypertension were significantly associated with an increased incidence of cataract.[23] Moreover, Jacques et al[6] also reported that women with a BMI of ≥30 had a higher prevalence of PSC opacities than women with a BMI of <25 (OR: 2.5).

We acknowledge that our study has some limitations. First, as this is a cross-sectional, case–control study, our ability to explore the exact mechanisms underlying the associations between serum lipid concentration and ARC remained limited. This in turn limited our ability to make exact causal inferences between serum lipid concentration and ARC. Second, the subjects involved in the study are from a racially homogeneous background (all Chinese), which may limit the generalisability of the results. Third, we did not consider treatment with lipid-lowering medications and how it might affect the results. Last, the methods used to analyse lipid levels in the blood are superficial. A more sophisticated gas chromatography-mass spectrometer (GC-MS) approach should be used in the future study. Therefore, a multicentre study with a larger sample size should be conducted in the future.

In conclusion, the findings from this case–control study suggest that the serum LDL-C, TG, CHO and APOA concentrations were significantly higher in the ARC patients than in the controls. The serum LDL-C and TG concentration were significantly associated with ARC. In addition, no significant difference in the serum lipid levels was found between the nuclear, cortical and PSC cataract groups. These findings indicate a need for health promotional activities aimed at controlling blood lipid concentration among high-risk populations.

**Contributors** SL, WC and MS were responsible for the conception and design of the manuscript. SL and DL analysed and interpreted the data. SL, YZ and JT collected the data. SL wrote the paper. All the authors critically reviewed the manuscript for scientific content and approved the final version. The authors thank all the participants and the staff (the doctors, nurses and administrative staff) at Ey and ENT Hospital, Shanghai Medical College, Fudan University for their contributions to this research.

**Funding** This research project was supported by the Shanghai Municipal Commission of Health and Family Planning (20174Y0169), Shanghai Sailing Program (18YF1403500).

**Competing interests** None declared.

**Patient consent** Obtained.

**Ethics approval** The ethics committee of the Eye and ENT Hospital, Shanghai Medical College approved this study.

**Provenance and peer review** Not commissioned; externally peer reviewed.

**Data sharing statement** No additional data are available.

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
