## [Reviewer comments · BMJ Open]

ARTICLE DETAILS

TITLE (PROVISIONAL)	Association between serum lipids concentration and patients with age-related cataract in China: a cross-sectional, case-control study
AUTHORS	Li, shengjie; li, dan; zhang, yudong; teng, jisen; shao, mingxi; cao, wenjun

VERSION 1 – REVIEW

REVIEWER	Hanna Lesiewska Collegium Medicum UMK, Poland
REVIEW RETURNED	28-Jan-2018

GENERAL COMMENTS	The article provides some interesting information on lipids concentration in cataract patients. Since some answers are still missing this study can be helpful. The authors have evaluated the group of Chinese age-related cataract patients and controls. The main limitation of that study is that it did not consider the lipid-lowering medications in groups studied, although the authors mentioned this problem. The conclusion of the abstract should be shorten; The serum LDL-C and TG levels were significantly higher in patients with ARC is included in the Results. The sentence “This is the first study focusing on the evaluation of the serum lipid concentration ...” should be removed as up till now there are lot of studies on this subject, some of them cited by authors. The statistics should be improved. As the collected data were not normally distributed (or were they?), the use of Student's t-test was invalid; the nonparametric test should be used. The data should be presented rather as medians instead of means. The term “vision” in table 1 should be replaced by “visual acuity”. Both figures (1 and 2) are unnecessary as they duplicate the content of table 2 and 3.
--

REVIEWER	Vivek Gupta Macquarie University , Australia
REVIEW RETURNED	09-Feb-2018

GENERAL COMMENTS	This is an overall well planned study and the manuscript is well written. Authors should clearly indicate the ethics approval number in the methodology. The effects of cataract associated or other disease treatment e.g; cardiac or diabetes related drugs on the systemic lipid profile is a major limitation of the study. Further the
--

	effects of diet and exercise/ life style has not been controlled and could be a major confounder. The effects of age associated disorders could further complicate the analysis which has not been considered. Authors should clearly indicate the age of the subjects that were included in the study and why this range was selected. Authors should plot a graph that could reflect correlation of individual subject's age with lipid profile. This will be a more objective representation of the data. The details of the methods used to analyse lipid levels in the blood are superficial. It will be highly valuable if the authors could screen the lipid profile using a more sophisticated GC-MS approach. At least, this limitation should also be discussed in the discussion section of the manuscript.
--	--

VERSION 1 – AUTHOR RESPONSE

Response to Editor:

1. The Editorial's comment: Please include the setting/country in the title.

The authors' answer: Thank you for your advice. The country has been added in the title.

2. The Editorial's comment: Please include a completed copy of the STROBE checklist, rather than the CONSORT checklist.

The authors' answer: A completed copy of the STROBE checklist for observational studies has been uploaded.

3. The Editorial's comment: The Strengths and Limitations section should just consist of points on the strengths and limitations of the study and study design. It should not present any results of the study.

The authors' answer: Thank you for your advice. Correction has been made in the revised version.

Response to Reviewer #1

1. The reviewer's comment: The article provides some interesting information on lipids concentration in cataract patients. Since some answers are still missing this study can be helpful. The authors have evaluated the group of Chinese age-related cataract patients and controls. The main limitation of that study is that it did not consider the lipid-lowering medications in groups studied, although the authors mentioned this problem.

The authors' answer: Thank you for your advice and we agree with your opinion. We did not consider treatment with lipid-lowering medications. Therefore, a study should be conducted to solve this limitation in the future.

2. The reviewer's comment: The conclusion of the abstract should be shorten; The serum LDL-C and TG levels were significantly higher in patients with ARC is included in the Results.

The authors' answer: Thank you for your advice. The conclusion of the abstract has been shorten in the revised version.

3. The reviewer's comment: The sentence "This is the first study focusing on the evaluation of the serum lipid concentration ..." should be removed as up till now there are lot of studies on this subject, some of them cited by authors.

The authors' answer: Thank you for your advice for making improvements on our manuscript. The sentence has been removed from the manuscripts.

4. The reviewer's comment: The statistics should be improved. As the collected data were not normally distributed (or were they?), the use of Student's t-test was invalid; the nonparametric test should be used. The data should be presented rather as medians instead of means.

The authors' answer: Thank you for your advice. Normality of the data was assessed with the Kolmogorov-Smirnoff test. The collected data were normally distributed.

	Kolmogorov-Smirnov value	P value
ARC group		
LDL-C	0.827	0.500
TG	1.178	0.124
HDL-C	0.631	0.820
APO A	0.803	0.539
APO B	0.923	0.362
CHO	0.743	0.638
Control group		
LDL-C	0.743	0.639
TG	1.050	0.221
HDL-C	1.065	0.206

APO A	0.788	0.564
APO B	0.905	0.386
CHO	0.733	0.655

5. The reviewer's comment: The term "vision" in table 1 should be replaced by "visual acuity".
The authors' answer: Thank you for your advice for making improvements on our manuscript.
Correction has been made in the revised version.

6. The reviewer's comment: Both figures (1 and 2) are unnecessary as they duplicate the content of table 2 and 3.
The authors' answer: Both figures (1 and 2) have been removed from the manuscripts.

Response to Reviewer #2

1. The reviewer's comment: This is an overall well planned study and the manuscript is well written. Authors should clearly indicate the ethics approval number in the methodology. The effects of cataract associated or other disease treatment e.g; cardiac or diabetes related drugs on the systemic lipid profile is a major limitation of the study. Further the effects of diet and exercise/ life style has not been controlled and could be a major confounder.

The authors' answer: Thank you for your advice and we agree with your opinion. (1) The ethics approval number (EENT2015011) has been added in the revised version. (2) We did not consider treatment with lipid-lowering medications is a major limitation of the study. (3) In this study, we conducted a face-to-face questionnaire interviews to gather data such as life habits, diet, tobacco and alcohol intake, and so on, but the effects of diet and exercise/ life style has not been controlled. As we all know, high-fat and meat diet habits and high level of serum lipids were closely related. We supposed that a high-fat and meat diet habits lead to high level of serum lipids, then high serum lipids concentration is significantly associated with ARC patients and may play an important role in the onset and development of ARC. Therefore, ARC patients with high-fat and meat diet habits should not be excluded from this study. A prospective, large sample study should be conducted to solve this problem in the future.

2. The reviewer's comment: The effects of age associated disorders could further complicate the analysis which has not been considered. Authors should clearly indicate the age of the subjects that were included in the study and why this range was selected.

The authors' answer: Thank you for your advice. Patients with ARC were consecutively recruited from the Department of Ophthalmology & Visual Science, Eye & ENT Hospital of Fudan University between June 2016 and June 2017. Normal controls were consecutively recruited from individuals who participated in annual health screenings during the study period. The age range of ARC group (46 to 85) and control group (48 to 79) has been added in the table 1. Tang et al. reported that the age range of ARC was 45 to 80+ in China [Tang, et al. IOVS. 2017], which was similar to our study. In this study, the age range of the subjects was not selected.

3. The reviewer's comment: Authors should plot a graph that could reflect correlation of individual subject's age with lipid profile. This will be a more objective representation of the data.

The authors' answer: Thank you for your advice. A graph which reflects correlation of individual subject's age with lipid profile has been added in the manuscripts (Supplementary Figure 1).

4. The reviewer's comment: The details of the methods used to analyse lipid levels in the blood are superficial. It will be highly valuable if the authors could screen the lipid profile using a more sophisticated GC-MS approach. At least, this limitation should also be discussed in the discussion section of the manuscript.

The authors' answer: Thank you for your advice for making improvements on our manuscript. This limitation has been added in the discussion section of the manuscript.

VERSION 2 – REVIEW

REVIEWER	Vivek Gupta Macquarie University, Australia
REVIEW RETURNED	27-Feb-2018

GENERAL COMMENTS	The authors have revised the manuscript and addressed the concerns that were raised.
--

REVIEWER	Hanna Lesiewska Collegium Medicum, Bydgoszcz, Nicolaus Copernicus University, Toruń, Poland
REVIEW RETURNED	04-Mar-2018

GENERAL COMMENTS	The manuscript has been improved by the authors and in my opinion can be accepted for the publication.
--